# Global Landscape Review of Serotype-Specific Invasive Pneumococcal Disease Surveillance among Countries Using PCV10/13: The Pneumococcal Serotype Replacement and Distribution Estimation (PSERENADE) Project

**DOI:** 10.3390/microorganisms9040742

**Published:** 2021-04-02

**Authors:** Maria Deloria Knoll, Julia C. Bennett, Maria Garcia Quesada, Eunice W. Kagucia, Meagan E. Peterson, Daniel R. Feikin, Adam L. Cohen, Marissa K. Hetrich, Yangyupei Yang, Jenna N. Sinkevitch, Krow Ampofo, Laurie Aukes, Sabrina Bacci, Godfrey Bigogo, Maria-Cristina C. Brandileone, Michael G. Bruce, Romina Camilli, Jesús Castilla, Guanhao Chan, Grettel Chanto Chacón, Pilar Ciruela, Heather Cook, Mary Corcoran, Ron Dagan, Kostas Danis, Sara de Miguel, Philippe De Wals, Stefanie Desmet, Yvonne Galloway, Theano Georgakopoulou, Laura L. Hammitt, Markus Hilty, Pak-Leung Ho, Sanjay Jayasinghe, James D. Kellner, Jackie Kleynhans, Mirjam J. Knol, Jana Kozakova, Karl Gústaf Kristinsson, Shamez N. Ladhani, Claudia S. Lara, Maria Eugenia León, Tiia Lepp, Grant A. Mackenzie, Lucia Mad’arová, Allison McGeer, Tuya Mungun, Jason M. Mwenda, J. Pekka Nuorti, Néhémie Nzoyikorera, Kazunori Oishi, Lucia Helena De Oliveira, Metka Paragi, Tamara Pilishvili, Rodrigo Puentes, Eric Rafai, Samir K. Saha, Larisa Savrasova, Camelia Savulescu, J. Anthony Scott, Kevin J. Scott, Fatima Serhan, Lena Petrova Setchanova, Nadja Sinkovec Zorko, Anna Skoczyńska, Todd D. Swarthout, Palle Valentiner-Branth, Mark van der Linden, Didrik F. Vestrheim, Anne von Gottberg, Inci Yildirim, Kyla Hayford

**Affiliations:** 1Johns Hopkins Bloomberg School of Public Health, Baltimore, MD 21205, USA; jbenne63@jhu.edu (J.C.B.); mgarci64@jhmi.edu (M.G.Q.); meaganepeterson@gmail.com (M.E.P.); mhetric2@jhmi.edu (M.K.H.); yyang165@jhmi.edu (Y.Y.); jsinkev1@jhu.edu (J.N.S.); lhammitt@jhu.edu (L.L.H.); kylahayford@jhu.edu (K.H.); 2Epidemiology and Demography Department, KEMRI-Wellcome Trust Research Programme, Centre for Geographic Medicine-Coast, P.O. Box 230-80108, Kilifi, Kenya; EKagucia@kemri-wellcome.org (E.W.K.); anthony.scott@lshtm.ac.uk (J.A.S.); 3Independent Consultant, 1296 Coppet, Switzerland; drf3217@gmail.com; 4World Health Organization, 1202 Geneva, Switzerland; dvj1@cdc.gov (A.L.C.); serhanfa@who.int (F.S.); 5Division of Pediatric Infectious Diseases, Department of Pediatrics, University of Utah Health Sciences Center, Salt Lake City, UT 84132, USA; Krow.Ampofo@hsc.utah.edu; 6Vaccine Study Center, Kaiser Permanente, Oakland, CA 94612, USA; Laurie.A.Aukes@kp.org; 7European Centre for Disease Prevention and Control, 169 73 Solna, Sweden; Sabrina.Bacci@ecdc.europa.eu; 8Centre for Global Health Research, Kenya Medical Research Institute, P.O. Box 1578-40100, Kisumu, Kenya; GBigogo@kemricdc.org; 9National Laboratory for Meningitis and Pneumococcal Infections, Center of Bacteriology, Institute Adolfo Lutz (IAL), São Paulo 01246-902, Brazil; maria.brandileone@ial.sp.gov.br; 10Arctic Investigations Program, Division of Preparedness and Emerging Infections, National Center for Emerging and Zoonotic Infectious Diseases, Centers for Disease Control and Prevention, Anchorage, AK 99508, USA; zwa8@cdc.gov; 11Department of Infectious Diseases, Italian National Institute of Health (Istituto Superiore di Sanità, ISS), 00161 Rome, Italy; romina.camilli@iss.it; 12CIBER Epidemiología y Salud Pública, (CIBERESP), 28029 Madrid, Spain; jcastilc@navarra.es (J.C.); pilar.ciruela@gencat.cat (P.C.); 13Instituto de Salud Pública de Navarra-IdiSNA, 31003 Pamplona, Spain; 14Singapore Ministry of Health, Communicable Diseases Division, Singapore 308442, Singapore; CHAN_Guanhao@moh.gov.sg; 15Instituto Costarricense de Investigación y Enseñanza en Nutrición y Salud, Tres Ríos, 30301 Cartago, Costa Rica; gchanto@inciensa.sa.cr; 16Surveillance and Public Health Emergency Response, Public Health Agency of Catalonia, 08005 Barcelona, Spain; 17Centre for Disease Control, Department of Health and Community Services, Darwin City, NT 8000, Australia; hcdarwin@gmail.com; 18Irish Meningitis and Sepsis Reference Laboratory, Children’s Health Ireland at Temple Street, Temple Street, D01 YC76 Dublin 1, Ireland; mary.corcoran@cuh.ie; 19The Faculty of Health Sciences, Ben-Gurion University of the Negev, 8410501 Beer-Sheva, Israel; rdagan@bgu.ac.il; 20Santé Publique France, the French National Public Health Agency, FR-94410 Saint Maurice, France; Costas.DANIS@santepubliquefrance.fr; 21Epidemiology Department, Dirección General de Salud Pública, 28009 Madrid, Spain; sarade.miguel@salud.madrid.org; 22Department of Social and Preventive Medicine, Laval University, Québec, QC G1V 0A6, Canada; philippe.dewals@criucpq.ulaval.ca; 23Department of Microbiology, Immunology and Transplantation, KU Leuven, 3000 Leuven, Belgium; stefanie.desmet@uzleuven.be; 24National Reference Centre for Streptococcus Pneumoniae, University Hospitals Leuven, 3000 Leuven, Belgium; 25Epidemiology Team, Institute of Environmental Science and Research, Porirua, Wellington 5022, New Zealand; Yvonne.Galloway@esr.cri.nz; 26National Public Health Organisation, 15123 Athens, Greece; t.georgakopoulou@eody.gov.gr; 27Swiss National Reference Centre for invasive Pneumococci, Institute for Infectious Diseases, University of Bern, 3012 Bern, Switzerland; Markus.Hilty@ifik.unibe.ch; 28Department of Microbiology and Carol Yu Centre for Infection, Queen Mary Hospital, The University of Hong Kong, Hong Kong, China; plho@hku.hk; 29National Centre for Immunisation Research and Surveillance and Discipline of Child and Adolescent Health, Children’s Hospital Westmead Clinical School, Faculty of Medicine and Health, University of Sydney, Westmead, NSW 2145, Australia; sanjay.jayasinghe@health.nsw.gov.au; 30Department of Pediatrics, University of Calgary, and Alberta Health Services, Calgary, AB T3B 6A8, Canada; kellner@ucalgary.ca; 31Centre for Respiratory Diseases and Meningitis, National Institute for Communicable Diseases of the National Health Laboratory Service, Sandringham, Johannesburg 2192, South Africa; JackieL@nicd.ac.za (J.K.); annev@nicd.ac.za (A.v.G.); 32School of Public Health, Faculty of Health Sciences, University of the Witwatersrand, Braamfontein, Johannesburg 2000, South Africa; 33National Institute for Public Health and the Environment, 3721 MA Bilthoven, The Netherlands; mirjam.knol@rivm.nl; 34National Institute of Public Health (NIPH), 100 42 Praha, Czech Republic; jana.kozakova@szu.cz; 35Department of Clinical Microbiology, Landspitali-The National University Hospital, Hringbraut, 101 Reykjavik, Iceland; karl@landspitali.is; 36Immunisation and Countermeasures Division, Public Health England, London NW9 5EQ, UK; shamez.ladhani@phe.gov.uk; 37Servicio de Bacteriología Clínica, Departamento de Bacteriología, INEI-ANLIS “Dr. Carlos G. Malbrán”, Buenos Aires C1282 AFF, Argentina; cslara@anlis.gob.ar; 38Laboratorio Central de Salud Pública, Asunción, Paraguay (Central Laboratory of Public Health, Asunción, Paraguay), Asunción, Paraguay; bacteriologia.lcsp@mspbs.gov.py; 39Department of Communicable Disease and Control and Health Protection, Public Health Agency of Sweden, 171 82 Solna, Sweden; tiia.lepp@folkhalsomyndigheten.se; 40Faculty of Infectious and Tropical Diseases, London School of Hygiene & Tropical Medicine, Keppel St, London WC1E 7HT, UK; gmackenzie@mrc.gm; 41Medical Research Council Unit the Gambia at London School of Hygiene & Tropical Medicine, P.O. Box 273, Banjul, The Gambia; 42New Vaccines Group, Murdoch Children’s Research Institute, Parkville, Melbourne, VIC 3052, Australia; 43National Reference Centre for Pneumococcal and Haemophilus Diseases, Regional Authority of Public Health, 975 56 Banská Bystrica, Slovakia; madarova@vzbb.sk; 44Toronto Invasive Bacterial Diseases Network, and Department of Laboratory Medicine and Pathobiology, University of Toronto, Toronto, ON M5S 1A8, Canada; Allison.McGeer@sinaihealth.ca; 45National Center of Communicable Diseases (NCCD), Ministry of Health, Bayanzurkh District, Ulaanbaatar 13336, Mongolia; tuya_mungun@yahoo.com; 46World Health Organization Regional Office for Africa, P.O. Box 06, Brazzaville, Congo; mwendaj@who.int; 47Department of Health Security, Finnish Institute for Health and Welfare, 00271 Helsinki, Finland; pekka.nuorti@tuni.fi; 48Health Sciences Unit, Faculty of Social Sciences, Tampere University, 33100 Tampere, Finland; 49Bacteriology-Virology and Hospital Hygiene Laboratory, Ibn Rochd University Hospital Centre, Casablanca 20250, Morocco; nzoyikorera@yahoo.fr; 50Department of Microbiology, Faculty of Medicine and Pharmacy, Hassan II University of Casablanca, Casablanca 20000, Morocco; 51Toyama Institute of Health, Imizu, Toyama 939-0363, Japan; toyamaeiken1@chic.ocn.ne.jp; 52Pan American Health Organization, World Health Organization, Washington, DC 20037, USA; oliveirl@paho.org; 53Centre for Medical Microbiology, National Laboratory of Health, Environment and Food, 2000 Maribor, Slovenia; metka.paragi@nlzoh.si; 54National Center for Immunizations and Respiratory Diseases, Centers for Disease Control and Prevention, Atlanta, GA 30333, USA; tpilishvili@cdc.gov; 55Instituto de Salud Pública de Chile, Santiago 7780050, Santiago Metropolitan, Chile; rpuentes@ispch.cl; 56Ministry of Health and Medical Services, Suva, Fiji; eric.rafai@govnet.gov.fj; 57Child Health Research Foundation, Dhaka 1207, Bangladesh; samirk.sks@gmail.com; 58Centre for Disease Prevention and Control of Latvia, 1005 Riga, Latvia; larisa.savrasova@spkc.gov.lv; 59Doctoral Studies Department, Riga Stradinš University, 1007 Riga, Latvia; 60Epidemiology Department, Epiconcept, 75012 Paris, France; c.savulescu@epiconcept.fr; 61Bacterial Respiratory Infection Service, Scottish Microbiology Reference Laboratory, NHS GG&C, Glasgow, G31 2ER, UK; kevin.scott@ggc.scot.nhs.uk; 62Department of Medical Microbiology, Medical University of Sofia, Faculty of Medicine, 1431 Sofia, Bulgaria; lenasetchanova@hotmail.com; 63Communicable Diseases Centre, National Institute of Public Health, 1000 Ljubljana, Slovenia; Nadja.Sinkovec-Zorko@nijz.si; 64National Reference Centre for Bacterial Meningitis, National Medicines Institute, 00-725 Warsaw, Poland; a.skoczynska@nil.gov.pl; 65Malawi-Liverpool-Wellcome Trust Clinical Research Programme, P.O. Box 30096, Chichiri, Blantyre, Malawi; tswarthout@mlw.mw; 66NIHR Global Health Research Unit on Mucosal Pathogens, Division of Infection and Immunity, UCL, Bloomsbury, London WC1E 6BT, UK; 67Infectious Disease Epidemiology and Prevention, Statens Serum Institut, DK-2300 Copenhagen, Denmark; pvb@ssi.dk; 68National Reference Center for Streptococci, Department of Medical Microbiology, University Hospital RWTH Aachen, 52074 Aachen, Germany; mlinden@ukaachen.de; 69Department of Infection Control and Vaccine, Norwegian Institute of Public Health, 0456 Oslo, Norway; didrik.frimann.vestrheim@fhi.no; 70School of Pathology, Faculty of Health Sciences, University of the Witwatersrand, Braamfontein, Johannesburg 2000, South Africa; 71Department of Pediatrics, Yale New Haven Children’s Hospital, New Haven, CT 06504, USA; inci.yildirim@yale.edu

**Keywords:** global, invasive pneumococcal disease, pneumococcal meningitis, surveillance, pneumococcal conjugate vaccines

## Abstract

Serotype-specific surveillance for invasive pneumococcal disease (IPD) is essential for assessing the impact of 10- and 13-valent pneumococcal conjugate vaccines (PCV10/13). The Pneumococcal Serotype Replacement and Distribution Estimation (PSERENADE) project aimed to evaluate the global evidence to estimate the impact of PCV10/13 by age, product, schedule, and syndrome. Here we systematically characterize and summarize the global landscape of routine serotype-specific IPD surveillance in PCV10/13-using countries and describe the subset that are included in PSERENADE. Of 138 countries using PCV10/13 as of 2018, we identified 109 with IPD surveillance systems, 76 of which met PSERENADE data collection eligibility criteria. PSERENADE received data from most (n = 63, 82.9%), yielding 240,639 post-PCV10/13 introduction IPD cases. Pediatric and adult surveillance was represented from all geographic regions but was limited from lower income and high-burden countries. In PSERENADE, 18 sites evaluated PCV10, 42 PCV13, and 17 both; 17 sites used a 3 + 0 schedule, 38 used 2 + 1, 13 used 3 + 1, and 9 used mixed schedules. With such a sizeable and generally representative dataset, PSERENADE will be able to conduct robust analyses to estimate PCV impact and inform policy at national and global levels regarding adult immunization, schedule, and product choice, including for higher valency PCVs on the horizon.

## 1. Introduction

*Streptococcus pneumoniae* is an important cause of morbidity and mortality globally, in both children and adults [1,2]. In 2007, the World Health Organization (WHO) first recommended including pneumococcal conjugate vaccines (PCV) in childhood immunization programs worldwide to prevent pneumococcal disease. WHO encouraged countries to implement surveillance of invasive pneumococcal disease (IPD) to establish a baseline rate of disease for evaluating vaccine impact [3]. In 2019, WHO expanded IPD surveillance recommendations to encourage high-quality sentinel surveillance to monitor the distribution of serotypes causing IPD and ideally population-based surveillance for evaluating PCV impact on IPD incidence and serotype replacement disease [4]. By 2020, 145 countries, including countries from all regions of the world, had introduced PCV into infant immunization programs [5], many of which have IPD surveillance systems [6,7,8,9,10]. However, an individual country’s ability to assess vaccine impact and inform policy can be limited by small sample size, limited years of available data either pre- or post-vaccine introduction, limited serotyping capacity, lack of a population catchment area for estimating incidence rates, changes in surveillance systems over time that bias inferences on vaccine impact, or insufficient characterization of cases or evaluation of the detection system to enable assessment of potential bias [11]. Further, unrelated events and temporal changes that influence health or access to care and natural fluctuations in pneumococcal serotypes over time may obscure PCV impact. Even sites not affected by these issues cannot assess the long-term relative merits across PCV products or schedules among both vaccinated and unvaccinated individuals, and their results may not be generalizable to other settings without robust data. Multi-site analyses that include data from many surveillance sites representing a variety of settings and PCV regimens can overcome these limitations. Multisite analyses also lead to greater understanding of pneumococcal epidemiology and PCV impact around the world, and where there is heterogeneity, to greater understanding of the factors driving it, e.g., differences in local epidemiology versus PCV use.

WHO’s Strategic Advisory Group of Experts (SAGE) on Immunization previously commissioned an analysis of PCV7 (Prevenar/Prevnar, Pfizer) impact [11] and several global and regional systematic reviews of IPD serotype distribution have also been conducted [12,13,14,15]. However, these reviews do not reflect the current setting of PCV10 (Synflorix, GlaxoSmithKline) and PCV13 (Prevenar13/Prevnar13, Pfizer) use, evaluate only published data, do not evaluate effects of PCV10 and PCV13 separately, or do not account for duration of PCV use. An updated, more comprehensive global analysis of the long-term effects of PCV10/13 on serotype-specific IPD incidence and serotype distribution is needed to inform policy related to pneumococcal epidemiology in PCV10/13-using countries, the potential value of future higher-valency PCVs, and global and national vaccination policy around product choice and schedule for children and immunization recommendations for adults.

WHO commissioned the Pneumococcal Serotype Replacement and Distribution Estimation (PSERENADE) project to summarize and estimate the impact of PCV10/13 programs on IPD incidence and serotype distribution among children and adults. Here we aimed to describe the landscape of available published and unpublished serotype-specific IPD surveillance data globally that can be used for evaluating vaccine impact, to identify limitations and gaps in the availability of IPD surveillance data globally, and to describe the surveillance sites included in PSERENADE to provide greater clarity in how the data used in PSERENADE analyses were gathered and processed.

## 2. Materials and Methods

### 2.1. Identification of Surveillance Sites

We aimed to systematically identify sites conducting serotype-specific IPD surveillance in countries where PCV10 or PCV13 was universally recommended for all infants by 1 January 2017 to ensure at least one full year of post-PCV10/13 surveillance data. Countries using PCV10/13 and their year of introduction were identified using View-Hub, a publicly available database with current information on PCV use worldwide [5]. IPD surveillance sites were identified using multiple approaches. First, we contacted the following surveillance networks: WHO-coordinated Global Invasive Bacterial Vaccine Preventable Disease (IB-VPD) Surveillance Network, the Pan American Health Organization (PAHO) Sistema de Redes de Vigilancia de los Agentes responsables de Neumonias y Meningitis (SIREVA) Network, the European Centre for Disease Prevention and Control *Streptococcus pneumoniae* Invasive Disease Network (SpIDnet), The European Surveillance System (ECDC), and the U.S. Centers for Disease Control and Prevention (CDC) Active Bacterial Core Surveillance (ABCs) system. Second, we conducted a systematic literature review including articles published in any language with publication dates between 1 January 2011 and 20 December 2018 to identify additional sites where serotype-specific IPD surveillance was conducted for at least a full year following PCV10/13 introduction. Seven databases (Embase (with Medline), PubMed, Web of Science (all databases), Global Index Medicus (including regional databases), Africa Wide Information, Global Health Database, and PASCAL) were searched using search terms modified for each database that were reviewed by a specialist librarian (Appendix A C). Third, results from the PCV Review of Impact Evidence (PRIME) literature review [16] and the View-Hub PCV10/13 impact study module database [5] were used to identify other sites and to validate the search terms to ensure relevant studies were captured. Two reviewers fluent in the language of the written report independently screened all studies and a third reviewer adjudicated disagreements. Fourth, we reviewed citations from a prior literature search on changes in IPD incidence after PCV7 introduction, which included studies published in 1994–2010 [11]. Fifth, International Symposium on Pneumococci and Pneumococcal Diseases (ISPPD) abstracts from 2012–2018 were reviewed. Finally, experts on pneumococcal disease surveillance suggested additional countries or sites not yet identified. 

### 2.2. Data Collection

Site investigators of identified surveillance sites and corresponding authors of studies identified in the literature review were contacted by email. Surveillance data were evaluated for suitability for inclusion in analyses of IPD serotype distribution and PCV impact on IPD incidence over time using standardized criteria intended to ensure comparability of methods and PCV uptake across sites (Table 1). Sites with suitable data were invited to participate in the PSERENADE project and contribute IPD surveillance data. IPD was defined as the isolation or detection of pneumococcus from a normally sterile site or detection of pneumococcus in cerebral spinal fluid (CSF) or pleural fluid using *lytA*-based PCR or antigen testing; pneumococcus detected in blood by PCR was not considered IPD given its low specificity [17]. Datasets provided by sites were preferentially used over data abstracted from literature in order to include the most up-to-date and comprehensive data available and to optimize the level of detail needed for planned analyses. Characterization of PSERENADE-eligible sites that chose not to participate in PSERENADE is based on descriptions in the published literature.

Surveillance sites shared annual serotype-specific IPD case data by age in either an individual case-level or aggregate format using a standardized template. Population-based denominators were provided where available. Prior to sharing, data were de-identified and anonymized per The US Health Insurance Portability and Accountability Act (HIPAA) and The European Union (EU) General Data Protection Regulation 2016/679 (GDPR). Data were stored on a secure database at Johns Hopkins University. Where possible, the following additional case characteristics were provided: hospitalized vs. outpatient status (for children under five years of age), HIV status, specimen type, and clinical syndrome (meningitis vs. pneumonia). For meningitis, two case definitions were used: confirmed positive CSF (CSF+) and site-defined clinical meningitis syndrome. Pneumonia cases were defined based on site-specific definitions. Characterization of non-pneumonia/non-meningitis IPD cases was not requested given limitations in data availability.

Site investigators also completed a questionnaire describing the site’s surveillance system and laboratory methods for detection of pneumococcus and serotyping of cases. The questionnaire requested information on the country’s pneumococcal immunization program, including annual immunization uptake estimates representative of the population under surveillance, PCV schedule, year of PCV introduction and product used (including use of PCV7 prior to introduction of PCV10/13), catch-up campaigns, and adult pneumococcal vaccination programs. We also abstracted WHO and UNICEF Estimates of National Immunization Coverage (WUENIC) for national uptake with three doses of PCV for all years of available surveillance data [21]. In the absence of evidence to the contrary, we assumed countries receiving funding from Gavi, the Vaccine Alliance to support PCV implementation did not have an adult pneumococcal vaccine program. 

For eligible PAHO countries participating in the SIREVA II surveillance network, the WHO-coordinated Global IB-VPD network facilitated data transfer for children under five years of age. For countries with additional data reported in SIREVA II reports beyond what was available in the WHO Global IB-VPD database, data for patients of all ages were abstracted from 2006–2016 (the last year of available data at the time of abstraction) by year, age group, and serotype [22]. Discrepancies in abstraction were adjudicated by a third reviewer (MGQ) fluent in Spanish. Colombia’s SIREVA II data were abstracted from a separate report published by the country, which included annual data through 2018 [23]. SIREVA II diagnostic and laboratory methods were abstracted from a standardized laboratory manual [24].

A standard data quality review was conducted independently for each site by two PSERENADE team members. Descriptive figures of the data with respect to each of the data quality check elements in Table 2 were shared with investigators with expertise in IPD surveillance at each site to assess the quality of the data. These characterizations and discussions with investigators at each site were used to define eligibility by year, age group, and syndrome for the various subsequent primary and secondary analyses of the study. 

PCV-using countries that had IPD surveillance data were summarized by data collection eligibility criteria and participation in PSERENADE. Sites were characterized by UN region [25], World Bank income level [26], under five mortality rate [27], childhood pneumococcal disease burden prior to PCV introduction [5], Gavi-eligibility status, PCV product, and PCV schedule. The surveillance systems and PCV programs were also described and summarized for sites included in PSERENADE.

## 3. Results

Pediatric and adult IPD surveillance data were available in every UN region of the world, representing countries from all World Bank income levels, under five mortality rate strata, levels of IPD disease burden, PCV products, and infant PCV schedules (Table 3). Of 138 countries with a universal infant PCV10/13 program operational for one or more years by January 1, 2018, we identified 109 conducting IPD surveillance (Table 3, Figure 1). Of these, 76 (69.7%) had surveillance that met PSERENADE eligibility criteria for data collection (Table 1) and 62 (81.6%) of those eligible participated. Surveillance sites in 14 countries that met data collection eligibility criteria did not contribute data to PSERENADE because they either did not respond or declined to participate. Characteristics associated with participation were not evaluated, but the proportion of participating eligible sites are detailed for each region (Table 3). The resulting dataset contained incidence rate data from 38 countries for evaluating PCV impact and case count data only from 24 additional countries for estimating serotype distribution. 

Eligibility of IPD surveillance data varied by region, income level, and epidemiological setting (Table 3). In Asia and Africa, where most pneumococcal deaths occur, fewer than half (48.3%) of the 58 countries conducting IPD surveillance met PSERENADE inclusion criteria, compared to 75.0–100% of countries elsewhere, and only 57.1% of the 28 that were eligible participated in PSERENADE. Although most (90.5%) PCV-using low-income countries (LICs) had IPD surveillance, the surveillance was less likely to meet eligibility criteria than that in upper-middle-(UMICs) or high-income countries (HICs) (47.4% for LICs vs. 78.9 for UMICs and 88.9% for HICs). Among those countries with surveillance meeting eligibility criteria, LICs were also less likely to contribute to PSERENADE (44.4% vs. 82.5–93.3%). Similarly, countries with high or medium under-5 mortality rates were less likely to have surveillance systems meeting eligibility criteria (38.5% and 44.0%, respectively) than low mortality countries (84.5%), and of the 13 high-mortality countries with IPD surveillance, only 5 (38.5%) were eligible for PSERENADE and only 2 participated, neither of which had population-based denominators to estimate incidence rates. There were 19 Gavi-eligible PCV-using countries with IPD surveillance eligible for PSERENADE, 13 (68.4%) of which participated, including 5 with incidence data. Of the 61 countries using a schedule with three primary doses and no booster (3 + 0), only 22 (36.1%) had eligible data, compared to 56 (70.0%) of 80 countries using an infant PCV schedule with a booster dose (3 + 1 or 2 + 1). Although the proportion of countries with surveillance systems meeting eligibility criteria was similar by PCV product (PCV13: 64.7%; PCV10: 71.4%), there were more PCV13-using countries eligible for PSERENADE analyses (n = 44 vs. 15).

Seventy-seven sites from 62 countries participated in PSERENADE (Table 3 and Table 4). All surveillance sites contributing data to PSERENADE collected pediatric data; although 88.0% overall also collected adult IPD data, those that did not were disproportionately from Sub-Saharan Africa and Asia where only 54.5% and 60.0% of sites, respectively, collected adult IPD data (Table 4). Data from the period prior to PCV introduction was available from 58 (77.3%) of surveillance sites. Although 51 (68.0%) sites conducted population-based surveillance with population denominators enabling incidence estimation, few of these were from the regions of Latin America and the Caribbean (three sites from two countries), Sub-Saharan Africa (six sites from four countries), and Northwestern Africa and Western Asia (two sites from two countries) (Table 4 and Appendix A).

All surveillance sites collected both blood and CSF except those in Sub-Saharan Africa, of which two (18.2%) collected blood only (Table 4); 68.9% of surveillance sites collected pleural fluid, with this proportion also lowest in Sub-Saharan Africa (2/11; 18.2%). Cases were characterized by clinical syndrome at 77.3% of sites overall, but those that did not characterize cases by clinical syndrome were disproportionately from the Latin America and the Caribbean region (11 of 19). Most surveillance sites (77.3%) used detection methods on CSF beyond culture (42.7% used antigen detection and 72.0% used nucleic acid detection). To identify the serotype, most (85.1%) sites used Quellung reaction and 73.0% used another method, primarily PCR (62.2%) and latex agglutination (29.7%) (Table 4 and Appendix A).

In total, PSERENADE collected data on over 240,000 post-PCV10/13 IPD cases, with the majority from Europe (n = 142,586) and North America (n = 37,187), but with a substantial number also from Latin America and the Caribbean (n = 20,609), Sub-Saharan Africa (n = 19,734), and Oceania (n = 13,038) (Table 3). The average number of annual cases post-PCV10/13 was lowest among Sub-Saharan Africa (median across sites = 10) and Latin America and the Caribbean (median = 50) compared to other regions (median range: 124–548). The number of cases per site in total was generally lower for sites without surveillance among all ages, those with smaller population catchment areas, and those with fewer years since PCV10/13 introduction (data not shown). The median number of surveillance years post-PCV10/13 across regions ranged from 4 (Asia) to 7 (North America, Europe and Northern Africa/Western Asia; Table 3).

Most (54.5%) PSERENADE sites used PCV13, 23.4% used PCV10 and 22.1% used both products concurrently or switched between products (Table 3 and Table 5). The majority of sites introduced PCV10/13 without a catch-up program (69.9%) and have a booster dose schedule (77.9%). PCV10/13 immunization coverage across the post-PCV10/13 period was high in most sites (mean uptake 87.9%, range 55–98%). The majority of sites have an adult pneumococcal vaccine program for polysaccharide vaccine (PPV23) and/or PCV13. Among these, 62.3% and 63.6% of sites recommend PPV23 for older adults and individuals at high risk for IPD, respectively, and 35.1% and 55.8%, respectively, recommend PCV13. Data on adult PPV23 and PCV13 uptake were available from 24 sites; 45.8% had >50% uptake (data not shown).

## 4. Discussion

As part of the PSERENADE project, the largest and most comprehensive global serotype-specific IPD database was compiled though a comprehensive and systematic search. All available serotype-specific IPD surveillance data in countries using PCV10/13 were identified and characterized to evaluate the global evidence to estimate the impact of PCV10/13 by age, product, schedule, and syndrome. IPD surveillance is recommended by the WHO [4] and nearly 80% of countries using PCV in 2018 had an active IPD surveillance system. Seventy percent of these met PSERENADE eligibility criteria for potential to evaluate PCV impact or post-PCV-era serotype distribution, and over half of the eligible countries had annual IPD incidence rate data. Eligible IPD surveillance data existed for both children and adults, in all regions of the world, from both PCV10- and PCV13-using countries, from countries with and without a booster dose schedule, and from all income and infant mortality rate strata. The majority of countries that met PSERENADE eligibility criteria were from HICs and used a booster dose. Although there were eligible data from at least 15 countries representing low- or lower-middle income countries (LMICs), 3 + 0 schedules, and the African or Asian regions, when restricted to analyses of incidence or stratified by product, data become very sparse for addressing some questions. While there are challenges in drawing inferences from observational surveillance data and some gaps remain, the breadth and depth of the data compiled by PSERENADE increase our capacity to address many questions.

This multisite database overcomes common IPD data limitations, including having too few cases or years of data available, temporal confounding, and changes of surveillance systems over time. It can facilitate more robust results with greater accuracy by observing trends at many sites, thus increasing confidence in interpretation and improving PCV policy relevance at both national and global levels. The PSERENADE collaboration will enable robust analyses to answer policy-relevant epidemiologic questions. These questions relate to how well PCVs performed in reducing vaccine-type disease, the magnitude of indirect effects of PCVs on vaccine-type disease in unimmunized older children and adults, the degree and heterogeneity of serotype replacement (i.e., where non-vaccine type disease increases as a result of reduction of vaccine-types) in all age groups, how these may differ by product or schedule, what optimizes the impact of available pneumococcal vaccines, and what the potential impacts of future higher valency PCVs in addressing residual IPD will be. Furthermore, serotype-specific IPD incidence rate data will allow for estimation of the effectiveness of PCV10/13 against important vaccine-type or -related serotypes, in particular serotypes 3, 6A, 6C, and 19A, and the magnitude of replacement disease due to specific non-vaccine serotypes.

Although there was representativeness across a wide array of settings, the quantity and depth of data at some sites were limited, reducing its usefulness for analyses. Resource-poor settings, which are often those with the highest disease burden, face many challenges. As a result, these were more likely to have the smallest IPD sample sizes and less well characterized cases. For example, resource-poor settings are more likely to have frequent use of antibiotics that limit detection of *S. pneumoniae* by bacterial culture, an inability to identify cases meeting the IPD case definition, a lack of capacity to perform more sensitive PCR-based tests of the CSF or using PCR serotyping methods that are limited in the number of serotypes they can identify (both of which may bias the serotype distribution [31,32]), or an inability to link laboratory results with clinical data. In addition, outbreak-prone settings can be overwhelmed during peak seasons with case management, and may not be able to keep up with specimen collection, testing, and reporting to national surveillance systems [33]. These challenges are reflected by a greater proportion of surveillance sites from LMICs not meeting PSERENADE data collection eligibility criteria. Further, some sites with eligible data were unable to contribute to the project due to a lack of data management resources. However, many of these surveillance sites are still able to serve local purposes, such as serving as sentinels to identify pockets of vaccine-type disease where immunization uptake may be suboptimal. Eligible sites from LMICs that participated in PSERENADE were also more likely to have small sample size compared to those from UMICs or HICs. Further, although most (88%) surveillance included adults in addition to children, surveillance of adult disease was less common in Africa (54%) and Asia (60%). Pneumococcal pneumonia and meningitis are common in adults and surveillance in this age group is important for assessing indirect effects of infant immunization, including replacement disease, particularly in the meningitis belt of Africa, where serotype 1 IPD outbreaks are common [33,34,35,36,37,38,39]. Improving the quality of existing systems in key high-burden settings could help address remaining questions and increase representation from all settings in global analyses.

The majority of PSERENADE sites were able to classify IPD cases by specimen type, thus enabling identification of pneumococcal meningitis cases (cases with detection of pneumococcus in CSF), which is important for understanding syndrome-specific vaccine impact. This is important in regions with a history of meningitis outbreaks, such as the African meningitis belt. However, only a small proportion of surveillance sites have laboratory data systematically linked to clinical data to allow characterization of cases by clinical diagnoses. Therefore, few PSERENADE sites were able to identify bacteremic pneumonia cases because blood cultures are also obtained for other non-pneumonia IPD syndromes. As a result, few surveillance sites can directly assess the relative PCV impact on meningitis versus bacteremic pneumonia. Countries within the African meningitis belt also tended to have less comprehensive surveillance systems, resulting in IPD being reported predominantly from meningitis cases [36,40,41] or having incomplete data on age or serotype [42]. Improving the availability and characterization of cases has direct relevance to WHO’s global roadmap for defeating meningitis by 2030, which set targets for vaccine-preventable meningitis surveillance, including for pneumococcal meningitis. In particular, the roadmap calls for strengthening of surveillance systems where a lack of laboratory capacity and resources for conducting surveillance hinder meningitis outbreak responses and provide data of limited quality to inform vaccine use and evaluate vaccine impact [43].

Despite investment in IPD surveillance globally, important gaps remain in the availability of IPD data needed for some assessments of PCV impact across diverse settings. The vast majority of IPD surveillance data are from HICs in Europe and North America. In these settings, pneumococcal epidemiology, serotype distribution, and disease burden differ from LMICs, which makes it difficult to be confident that global analyses are fully representative. Africa and Asia are the most under-represented. Most sites in these regions report lower case counts despite having a higher disease burden. Furthermore, stable population-based surveillance over time, important for estimating incidence rates, is particularly sparse in LMICs using a 3 + 0 schedule; only 5 countries (Bangladesh, Fiji, The Gambia, Kenya, and Malawi) provided data, with Bangladesh providing data for children only. A limitation of PSERENADE was not being able to get all key data that are available, including from two important countries using 3 + 0 schedules and conducting surveillance in the meningitis belt, Ghana and Burkina Faso. Population-based surveillance data over time for estimating impact were particularly limited from Latin America and the Caribbean, where only Chile had population-based surveillance data for all IPD, and Brazil had population-based surveillance data for pneumococcal meningitis only. Currently, the ability to answer important sub-group questions, including the indirect effects and some serotype-specific effects of 3 + 0 schedules in high-burden settings (particularly in meningitis outbreak prone settings), is limited. Additional high-quality and well-characterized data for all ages in these settings will be needed to answer these questions.

## 5. Conclusions

A large amount of IPD data is available globally for both children and adults. PSERENADE’s systematic assessment and combined database of the available serotype-specific IPD surveillance data in countries using PCV10/13 will facilitate answering important questions as well as highlight the gaps needed to be filled to address remaining questions. These data have the potential to inform policy around pneumococcal vaccine use, for both PCV and PPV, at national and global levels, including recommendations concerning product choice, schedule, and adult immunization. The PSERENADE project has informed WHO SAGE recommendations around pneumococcal vaccine use in adults and in community outbreak settings [33] and will contribute important evidence for other pneumococcal vaccine policy decisions. The ongoing collection of serotype-specific IPD surveillance data in countries that have introduced or plan to introduce PCV, as recommended by WHO, will provide data needed to understand PCV impact and inform pneumococcal vaccine policy decisions, particularly if efforts are made to support and enhance surveillance capacity in key areas underrepresented in global analyses.

## Figures and Tables

**Figure 1 microorganisms-09-00742-f001:**
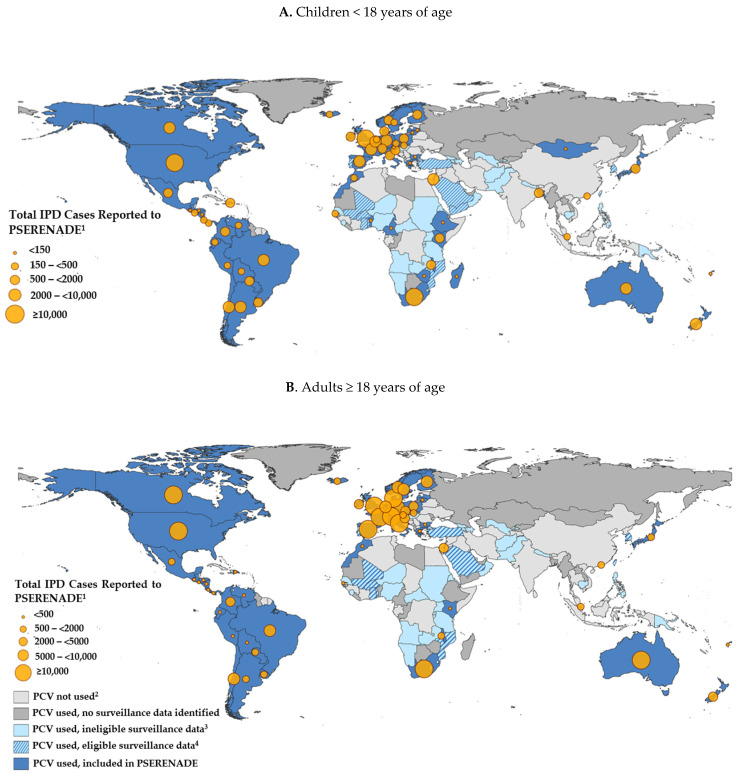
Availability of IPD surveillance data for countries with universal recommendations for PCV in the infant immunization program. ^1^ Cases from multiple surveillance sites within the same country were aggregated.^2^ PCV not universally introduced into the routine infant immunization program by 2018 (includes India which began sub-national introduction in 2017). ^3^ IPD surveillance data did not meet PSERENADE data collection eligibility criteria (Box 1). ^4^ IPD surveillance data met data collection eligibility criteria but did not participate in PSERENADE.

**Table 1 microorganisms-09-00742-t001:** Data collection inclusion criteria.

Data Collection Inclusion Criterion	Rationale
1. Site reports annual serotype-specific and age-specific IPD case counts obtained from normally sterile sites	Data must meet a standardized case definition of IPD to ensure comparability across sites, be stratified by age in order to evaluate direct and indirect effects, and be characterized by serotype in order to estimate serotype distributions and evaluate vaccine-type and serotype-specific changes in disease rates over time.
2. At least 50% of isolates serotyped per year in at least one age stratum	A minimum proportion of isolates must be serotyped to limit risks of non-representativeness of serotyped isolates, i.e., to ensure the serotype distribution based only on serotyped cases is not biased and to minimize chance from selective testing.
3. At least one complete year of data post-PCV10/13, excluding the year of introduction	Twelve continuous months are required to ensure data are not limited to an outbreak period and to control for seasonal fluctuations in disease or serotype-specific distribution.
4. At least 50% uptake for the primary PCV series at 12 months of age in at least one year post-PCV10/13	The goal is to evaluate PCV, not the immunization program. Therefore, vaccine uptake must be high enough to be able to affect serotype distribution/IPD incidence rates at the population level [18,19] and to represent the experience of countries with high coverage. It also serves to help eliminate heterogeneous results due to coverage to enable focus on effects of product and schedule.
5. Testing or reporting not limited to immunocompromised individuals or other specialized populations	PCV impact and serotype distribution may be different in specialized populations (e.g., HIV-positive populations) and may not be representative of the wider population [20].
6. No major changes or biases in surveillance that would affect estimates of serotype-specific proportions or rates	Changes in the surveillance system over the analysis period, such as a change in indication for blood culturing, introduction of new serotyping methods, or change in the population under surveillance, may bias interpretations of changes in incidence rates making it difficult to distinguish PCV effects from a change in the system. If changes are correlated with vaccine introduction, results may be incorrectly attributed to vaccine program impact.

**Table 2 microorganisms-09-00742-t002:** PSERENADE standard data quality review.

Data Quality Check	Rationale
A. Are there dramatic changes in overall IPD incidence rates (IR) from year to year that might not be explained by vaccine introduction?	Stable surveillance system, population structure and clinical practices should not exhibit dramatic unexplained changes.
B. Are vaccine-serotype IRs decreasing in the target age groups after vaccine introduction as expected?	Vaccine-serotype IRs should be decreasing in target age groups after vaccine introduction, given sufficient vaccine uptake.
C. Are there dramatic changes in overall case counts from year to year that might not be explained by vaccine introduction?	Dramatic unexplained changes in case counts could indicate changes in the surveillance system or clinical practices.
D. Are vaccine-serotype case counts decreasing in the target age groups after vaccine introduction as expected?	Vaccine-serotype case counts should be decreasing in target age groups after vaccine introduction, given sufficient vaccine uptake.
E. Have the number of cases due to serotype 14 and 6B among children < 5 years been eliminated or greatly reduced in the post-PCV era?	Serotype 14 and 6B should be decreasing after vaccine introduction. Persistent serotype 14 or 6B cases may indicate low immunization coverage or surveillance system changes.
F. Do the denominators used to calculate IRs in each age group change over time?	Population-based denominators should vary slightly but not substantially over time. If annual population denominators are not available (i.e., denominator only available in some years) rates may be an under- or over-estimate.
G. Do the denominators in each age group make sense relative to each other?	Based on conventional population age structures, we expect the number of children aged < 5 years to be less than adults aged ≥ 18 years. The number of adults aged ≥ 65 years would be expected to be less than that of adults aged 18–64 years.
H. Do all IPD IRs in each age group make sense relative to each other and the setting?	Expect IPD IRs to be highest in young children and older adults who are most vulnerable, but there can be exceptions in some settings where other age groups have age-associated excess risk [28].
I. Do at least 50% of cases for each age group/surveillance year stratum have a known serotype?	Ensures that the serotype distribution of serotyped cases is not biased or different from the serotype distribution of cases that were not serotyped or not fully serotyped. An exception can be made if the specimens were randomly selected for serotyping, when costs may prohibit all serotyped.
J. Does the site distinguish between: Serotype 6A and serotype 6C cases? Serotype 6B and serotype 6D cases?	In 2007 researchers discovered that pneumococci classified as serotype 6A on the basis of phenotype could be further distinguished chemically, resulting in identification of a new serotype, 6C [29]. Similarly, in 2009 serotype 6D was discovered as a chemically distinct serotype from 6B [30]. Pneumococci previously classified as serotype 6A or 6B would have to be retrospectively reevaluated to distinguish serotypes 6C and 6D, respectively.
K. Are undistinguished PCV13-type serotypes identifiable (e.g., ‘6A/6C’ instead of ‘6A’)?	Because undistinguished PCV13-type cases (e.g., 6A/6C) will need to be reapportioned based on the distribution of fully serotyped PCV13-type cases, confirmed ‘6A’ cases need to be differentiated from unconfirmed (i.e., might be 6C). Dates of changes in serotyping methods or documentation of retrospective reclassification efforts are required.

**Table 3 microorganisms-09-00742-t003:** Availability of invasive pneumococcal disease (IPD) surveillance data globally in PCV10/13-using countries.

Strata	Category	All PCV-Using Countries ^1^			Data in PSERENADE					
		A. Countries Using PCV, N (% of Countries ^2^)	B. PCV-Using Countries with IPD Surveillance, N (% of A ^2^)	C. Countries Eligible for PSERENADE, N (% of B ^2,3^)	D. Countries in PSERENADE, N (% of C ^2^)	E. Countries with Incidence Data ^4^, N (% of D ^2^)	F. Number of Surveillance Sites	G. Total Number of Cases in Post-PCV10/13 Years ^5^	H. Annual Number of Cases Averaged across Post-PCV10/13 Years, Median (IQR) ^5^^,6^	I. Number of Years Post-PCV10/13 with Data, Median (IQR) ^5^
Total	Total	138 (70.4%)	109 (79.0%)	76 (69.7%)	62 (81.6%)	38 (61.3%)	77	241,442	117 (26, 513)	7 (5, 7)
Region ^7^	North America	2 (100.0%)	2 (100.0%)	2 (100.0%)	2 (100.0%)	2 (100.0%)	10	37,187	124 (55, 269)	7 (7, 8)
	Latin America and the Caribbean	22 (66.7%)	19 (86.4%)	19 (100.0%)	18 (94.7%)	2 (11.1%)	19	20,609	50 (21, 227)	5 (4, 6)
	Europe	31 (73.8%)	26 (83.9%)	24 (92.3%)	23 (95.8%)	20 (87.0%)	26	142,586	548 (115, 918)	7 (6, 8)
	Sub-Saharan Africa	39 (81.2%)	31 (79.5%)	14 (45.2%)	9 (64.3%)	4 (44.4%)	11	19,734	10 (7, 23)	6 (5, 6)
	Northern Africa and Western Asia	17 (73.9%)	14 (82.4%)	7 (50.0%)	2 (28.6%)	2 (100.0%)	2	4,380	313 (171, 454)	7 (7, 7)
	Asia	17 (53.1%)	13 (76.5%)	7 (53.8%)	5 (71.4%)	5 (100.0%)	5	3,908	126 (77, 179)	4 (3, 7)
	Oceania	10 (62.5%)	4 (40.0%)	3 (75.0%)	3 (100.0%)	3 (100.0%)	4	13,038	274 (48, 748)	6 (6, 7)
World Bank Income level ^8^	High income	52 (83.9%)	45 (86.5%)	40 (88.9%)	33 (82.5%)	29 (87.9%)	46	206,562	377 (99, 824)	7 (6, 8)
	Upper middle income	27 (50.0%)	19 (70.4%)	15 (78.9%)	14 (93.3%)	3 (21.4%)	14	33,085	55 (21, 272)	6 (4, 7)
	Lower middle income	38 (74.5%)	26 (68.4%)	12 (46.2%)	11 (91.7%)	4 (36.4%)	12	968	12 (8, 19)	5 (4, 6)
	Low income	21 (72.4%)	19 (90.5%)	9 (47.4%)	4 (44.4%)	2 (50.0%)	5	827	10 (9, 29)	6 (5, 6)
Under 5 mortality rate (2018) ^9^	Low	87 (66.4%)	71 (81.6%)	60 (84.5%)	52 (86.7%)	33 (63.5%)	65	221,478	179 (42, 587)	7 (5, 7)
	Medium	35 (79.5%)	25 (71.4%)	11 (44.0%)	8 (72.7%)	5 (62.5%)	10	19,948	14 (9, 65)	6 (5, 7)
	High	16 (76.2%)	13 (81.2%)	5 (38.5%)	2 (40.0%)	0 (0.0%)	2	16	3 (3, 3)	6 (6, 6)
Pre-PCV under 5 Spn disease burden (2000) ^10^	Low burden	42 (77.8%)	38 (90.5%)	33 (86.8%)	29 (87.9%)	27 (93.1%)	41	200,066	469 (117, 878)	7 (7, 8)
	Medium burden	34 (60.7%)	23 (67.6%)	20 (87.0%)	17 (85.0%)	3 (17.6%)	18	20,356	64 (24, 275)	6 (5, 7)
	High burden	62 (74.7%)	48 (77.4%)	23 (47.9%)	16 (69.6%)	8 (50.0%)	18	21,020	17 (9, 30)	5 (4, 6)
Gavi status ^11^	Gavi	57 (78.1%)	44 (77.2%)	19 (43.2%)	13 (68.4%)	5 (38.5%)	15	1455	10 (7, 15)	5 (4, 6)
	Non-Gavi	81 (65.9%)	65 (80.2%)	57 (87.7%)	49 (86.0%)	33 (67.3%)	62	239,987	262 (57, 625)	7 (6, 8)
Product	PCV10	22 (15.9%) ^15^	21 (95.5%)	15 (71.4%)	14 (93.3%)	8 (57.1%)	18	23,967	49 (14, 416)	6 (5, 7)
	PCV13	93 (67.4%) ^15^	68 (73.1%)	44 (64.7%)	34 (77.3%)	19 (55.9%)	42	183,610	123 (31, 594)	7 (6, 7)
	PCV10 and PCV13 ^12^	23 (16.7%) ^15^	20 (87.0%)	17 (85.0%)	14 (82.4%)	11 (78.6%)	17	33,865	209 (56, 386)	7 (6, 7)
Schedule ^13^	3 + 0	58 (42.0%) ^16^	44 (75.9%)	20 (45.5%)	14 (70.0%)	5 (35.7%)	17	10,825	12 (8, 29)	6 (5, 6)
	2 + 1	48 (34.8%) ^16^	40 (83.3%)	36 (90.0%)	33 (91.7%)	20 (60.6%)	38	151,942	308 (70, 594)	7 (5, 7)
	3 + 1	19 (13.8%) ^16^	13 (68.4%)	10 (76.9%)	6 (60.0%)	5 (83.3%)	13	32,716	92 (42, 247)	7 (7, 8)
	3 + 0 and 2 + 1/3 + 1 ^14^	3 (2.2%) ^16^	2 (66.7%)	2 (100.0%)	1 (50.0%)	1 (100.0%)	0^14^	0	0 (0, 0)	0 (0, 0)
	3 + 1 and 2 + 1 ^15^	10 (7.2%) ^16^	10 (100.0%)	8 (80.0%)	8 (100.0%)	7 (87.5%)	9	45,959	634 (276, 932)	7 (7, 8)

^1^ Countries with a full year of a PCV10/13 immunization program for infants by 2018 (i.e., introduced by 1 January 2017). Countries with only a risk immunization program rather than universal are also not counted as PCV-using countries. Data from View-Hub [5]. Taiwan and Hong Kong are not merged with China in this table given differences in PCV use and availability of IPD surveillance data compared to the rest of China. ^2^ Percentage by category unless otherwise specified. ^3^ To be eligible for PSERENADE, a surveillance site must have had at least one full year of post-PCV10/13 IPD incidence or four years of post-PCV10/13 IPD case counts, over 50% vaccination uptake, and over 50% of cases serotyped by age/year group (Table 1). ^4^ Incidence data are only available for pneumococcal meningitis cases in Brazil and Greece, as opposed to all IPD in all other countries. ^5^ Post-PCV10/13 years exclude the year of introduction. ^6^ The average number of cases in post-PCV10/13 years was calculated for each surveillance site and used to estimate the median (IQR) across strata categories. ^7^ United Nations (UN) regions adapted from UN Statistics Division [25]. ^8^ World Bank Income level as of November 2020 [26]. ^9^ Under 5-year mortality rate data from United Nations Interagency Group for Child Mortality Estimation (2020), 2018 estimate by country. Low: <30 deaths per 1000 livebirths, medium: 30 to <75 deaths, high: 75 to <150 deaths [27]. ^10^ Pre-PCV pneumococcal disease burden estimates for children <5 years calculated as the sum of estimated pneumonia, meningitis, and invasive non-pneumonia, non-meningitis incidence rates in 2000 [5]. Strata were defined as fewer than 300 cases per 100,000 children (low burden), 300 to fewer than 2000 cases per 100,000 children (medium burden), or 2000 or more cases per 100,000 children (high burden). Countries missing any or all incidence rates were categorized as “Unknown”. ^11^ Gavi countries are those that are eligible or have graduated. ^12^ Countries that either used both products concurrently or switched between PCV10 and PCV13. ^13^ 3 + 0: three primary doses and no booster; 2 + 1: two primary doses and a booster; 3 + 1: three primary doses and a booster. ^14^ Countries that used PCV10/13 schedules with and without a booster dose. Australia, included in PSERENADE, uses 3 + 1 among indigenous populations and used 3 + 0 among non-indigenous populations until 2018, when non-indigenous changed to 2 + 1. Because Australia (non-indigenous) predominantly used 3 + 0 during the years described here, that surveillance site was categorized as 3 + 0 in columns FI, and Australia (indigenous) was categorized as a 3 + 1 surveillance site in columns F–I. Not included in PSERENADE were Trinidad and Tobago (switched from 3 + 0 to 3 + 1) and Libyan Arab Jamahiriya (switched from 3 + 0 to 2 + 1). ^15^ Countries that used 3 + 1 and 2 + 1 PCV10/13 schedules. All switched from 3 + 1 to 2 + 1 except for Poland, which uses 2 + 1 in the National Immunization Program (NIP) and 3 + 1 in the private market, and Canada, which uses 3 + 1 and/or 2 + 1 in different provinces. Canadian surveillance sites for individual provinces are categorized accordingly in columns F-I. ^16^ Percentage is of the 138 PCV-using countries.

**Table 4 microorganisms-09-00742-t004:** Summary of PSERENADE surveillance sites by region ^1,2.^

	North America N = 9	Latin America and the Caribbean N = 19	Europe N = 26	Sub-Saharan Africa N = 11	N. Africa and W. Asia N = 2	Asia N = 5	Oceania N = 3	Total N = 75
Availability of data, N (%)								
0–17 years	9 (100%)	19 (100%)	26 (100%)	11 (100%)	2 (100%)	5 (100%)	3 (100%)	75 (100%)
≥18 years	7 (77.8%)	19 (100%)	26 (100%)	6 (54.5%)	2 (100%)	3 (60.0%)	3 (100%)	66 (88.0%)
Pre-PCV period	7 (77.8%)	19 (100%)	17 (65.4%)	6 (54.5%)	2 (100%)	4 (80.0%)	3 (100%)	58 (77.3%)
PCV7 period ^3^	8 (88.9%)	8 (100%)	16 (84.2%)	2 (100%)	1 (100%)	3 (75.0%)	2 (100%)	40 (88.9%)
PCV10/13 period	9 (100%)	19 (100%)	26 (100%)	11 (100%)	2 (100%)	5 (100%)	3 (100%)	75 (100%)
Incidence data ^4^	9 (100%)	3 (15.8%)	23 (88.5%)	6 (54.5%)	2 (100%)	5 (100%)	3 (100%)	51 (68.0%)
Clinical syndrome data	8 (88.9%)	11 (57.9%)	20 (76.9%)	10 (90.9%)	1 (50.0%)	5 (100%)	3 (100%)	58 (77.3%)
Specimens collected, N (%) ^5^								
Blood	9 (100%)	19 (100%)	25 (100%)	11 (100%)	2 (100%)	5 (100%)	3 (100%)	74 (100%)
CSF	9 (100%)	19 (100%)	25 (100%)	9 (81.8%)	2 (100%)	5 (100%)	3 (100%)	72 (97.3%)
Pleural fluid	7 (77.8%)	18 (94.7%)	17 (68.0%)	2 (18.2%)	1 (50.0%)	4 (80.0%)	2 (66.7%)	51 (68.9%)
Additional detection methods, N (%)								
Nucleic acid	2 (22.2%)	16 (84.2%)	23 (88.5%)	6 (54.5%)	1 (50.0%)	4 (80.0%)	2 (66.7%)	54 (72.0%)
Antigen detection	0 (0.0%)	13 (68.4%)	14 (53.8%)	0 (0.0%)	0 (0.0%)	3 (60.0%)	2 (66.7%)	32 (42.7%)
Serotyping methods, N (%) ^5^								
Quellung	9 (100%)	19 (100%)	23 (92.0%)	4 (36.4%)	2 (100%)	3 (60.0%)	3 (100%)	63 (85.1%)
Non-Quellung	2 (22.2%)	12 (63.2%)	22 (88.0%)	11 (100%)	1 (50.0%)	4 (80.0%)	2 (66.7%)	54 (73.0%)
Latex agglutination	1 (11.1%)	2 (10.5%)	15 (60.0%)	3 (27.3%)	1 (50.0%)	0 (0.0%)	0 (0.0%)	22 (29.7%)
Any PCR method ^6^	2 (22.2%)	12 (63.2%)	14 (56.0%)	11 (100%)	1 (50.0%)	4 (80.0%)	2 (66.7%)	46 (62.2%)
PCR35/37/38 ^7,8^	0 (0.0%)	10 (52.6%)	2 (8.0%)	2 (18.2%)	0 (0.0%)	1 (20.0%)	0 (0.0%)	15 (20.3%)
PCR70/76 ^7,8^	2 (22.2%)	5 (26.3%)	8 (32.0%)	0 (0.0%)	1 (50.0%)	3 (60.0%)	1 (33.3%)	20 (27.0%)
Other method ^9^	1 (11.1%)	0 (0.0%)	6 (24.0%)	0 (0.0%)	0 (0.0%)	0 (0.0%)	0 (0.0%)	7 (9.5%)

^1^ Subpopulations (e.g., indigenous and non-indigenous) from the same surveillance system were presented as one site. Countries with more than one surveillance site are represented more than once. Data for individual surveillance sites are in Appendix A. ^2^ United Nations (UN) regions adapted from UN Statistics Division [25]. N. Africa and W. Asia: Northern Africa and Western Asia. ^3^ Sites that did not use PCV7 were excluded from calculations of PCV7 period data availability (not applicable). Total calculations are out of the 45 sites that used PCV7. ^4^ Incidence data from Brazil and Greece are for pneumococcal meningitis only. ^5^ One site (Lithuania) with unknown specimen type and serotyping data was excluded from calculations. Total calculations are out of 74 sites. ^6^ Comprised of sites that use PCR at any capacity—including those with unknown or custom PCR schemes that do not fall into PCR35/37/38 or PCR70/76 categories. ^7^ The number following “PCR” indicates the number of serotypes able to be identified by PCR. Similar serotyping capacities were grouped together. ^8^ Argentina, Mexico, and Paraguay use both PCR37 and PCR70 and are counted in both of those categories. ^9^ Includes sites that reported other serotyping methods: Whole genome sequencing (WGS), Next generation sequencing (NGS), Capsular sequence typing (CST), or Gel diffusion (GD).

**Table 5 microorganisms-09-00742-t005:** Description of infant and adult pneumococcal immunization use for PSERENADE sites.

Infant PCV Product ^1^	PCV10/13 Schedule ^1^	Region ^2^	Site ^3^	PCV7 Period	PCV10 Period	PCV13 Period	PCV10/13 Catch-Up	Mean PCV10/13 Uptake (%)		Other PCV/PPV Recommendations ^7^	
								Primary Series ^5^	WUENIC PCV3 ^6^	Adult	High Risk
PCV10	3 + 0	LA and C	Ecuador (SIREVA, WHO)	2010–2011	2010–	--	N	--	85	--	--
		Sub-Saharan Africa	Ethiopia (WHO)	-- ^4^	2011–	--	N	--	58	--	--
			Kenya, Asembo	--	2011–	--	Y	86	81	--	--
			Kenya, Kibera	--	2011–	--	N	87	81	--	--
			Kenya, Kilifi	--	2011–	--	Y	83	78	--	--
			Madagascar (WHO)	--	2012–	--	N	--	74	--	--
		Asia	Bangladesh	--	2015–	--	N	--	97	--	--
		Oceania	Fiji	--	2012–	--	N	90	99	--	--
	2 + 1	LA and C	Colombia (SIREVA)	--	2011–	--	N	--	82	--	PPV
		Europe	Austria (ECDC)^10^	--	2012–	--	N	--	--	PPV, PCV	PPV
			Finland	--	2010–	--	N	95	90	PPV, PCV	PPV, PCV
			Iceland	--	2011–	--	N	--	89	PPV	PPV, PCV
			Latvia	2010–2011	2012–	--	N	91	85	--	--
			Lithuania (ECDC)	--	2014–	--	Unk	--	82	--	--
			Slovenia	--	2015–2019	2019–	N	55 ^12^	55	PPV, PCV	PPV, PCV
	3 + 1	Europe	Bulgaria	--	2010–	--	N	--	91	PPV, PCV	--
	3 + 1/2 + 1	LA and C	Brazil	--	2010–	--	Y	91 ^8^	88	--	PPV, PCV
		Europe	Netherlands	2006–2011	2011–	--	N	95	94	--	PPV, PCV
PCV13	3 + 0	LA and C	Bolivia (SIREVA)	--	--	2014–	N	--	--	--	--
			Honduras (SIREVA, WHO)	--	--	2011–	N	--	99	--	--
			Nicaragua (SIREVA, WHO)	--	--	2010–	N	--	98	PPV	PPV
		Sub-Saharan Africa	Benin (WHO)	--	--	2011–	N	--	73	--	--
			Cameroon (WHO)	--	--	2011–	N	--	72	--	--
			Malawi, Blantyre District	--	--	2011–	Y	92	88	--	--
			The Gambia, Basse	2009–2011	--	2011–	N	77	95	--	--
			Zimbabwe (WHO)	--	--	2012–	N	--	90	--	--
	2 + 1	N. Am.	Canada, Alberta	2002–2010	--	2010–	N	88 ^8^	77	PPV	PPV, PCV
		LA and C	Argentina	--	--	2012–	Y	--	84	PPV, PCV	PPV, PCV
			Costa Rica	--	--	2011–	N	94	92	--	PPV, PCV
			Dominican Republic (SIREVA)	--	--	2013–	N	--	--	--	--
			Guatemala (SIREVA)	--	--	2012–	N	--	81	--	--
			Mexico (SIREVA)	2009–2013	--	2011–	Y	--	90	PPV	PPV
			Panama (SIREVA)	2010–2011	--	2011–	Unk	--	93	PPV, PCV	PPV, PCV
			Uruguay (SIREVA)	2008–2010	--	2010–	Y	--	94	PPV	--
		Europe	Denmark	2007–2010	--	2010–	N	91 ^8^	93	PPV, PCV	PPV, PCV
			France	2006–2010	--	2010–	N	93	90	--	PPV, PCV
			Ireland	2008–2010	--	2010–	N	91	91	PPV	PPV, PCV
			Italy	2006–2009	--	2010–	N	86 ^8^	87	PPV, PCV	--
			Norway	2006–2011	--	2011–	N	93	93	PPV	PPV, PCV
			Spain, Madrid	2006–2010	--	2010–	N	98	93	PPV, PCV	PPV, PCV
			Switzerland	2005–2010	--	2010–	Y	79 ^8^	77	--	PCV
			UK, England	2006–2009	--	2010–	N	94	92	PPV	PPV
			UK, Scotland	2006–2010	--	2010–	N	97	92	PPV	PPV, PCV
		N. Africa and W. Asia	Israel	2009–2010	--	2010–	N	95	93	PPV, PCV	PPV, PCV
		Sub-Saharan Africa	South Africa	2009–2011	--	2011–	Y	77 ^8^	77	PPV, PCV	PPV, PCV
		Asia	Mongolia	--	--	2016–	Y	93	20	--	--
			Singapore	2009–2011	--	2011–	Y	83	74	PPV, PCV	PPV, PCV
	3 + 1	N. Am.	USA, ABCs	2000–2009	--	2010–	Y	88	93	PPV, PCV	PPV, PCV
			USA, Alaska	2001–2009	--	2010–	Y	83	93	PPV, PCV	PPV, PCV
			USA, California	2000–2009	--	2010–	Y	96	93	PPV, PCV	PPV, PCV
			USA, Massachusetts	2000–2009	--	2010–	Y	94	93	PPV, PCV	PPV, PCV
			USA, Southwest (Indigenous)	2000–2009	--	2010–	Y	82	93	PPV, PCV	PPV, PCV
			USA, Utah	2000–2009	--	2010–	Y	88	93	PPV, PCV	PPV, PCV
		Europe	Greece ^10^	2006–2009	--	2010–	N	82	75	PPV, PCV	PPV, PCV
		Asia	Japan	2010–2013	--	2013–	N	94 ^8^	98	PPV	PPV
	3 + 0/2 + 1	Oceania	Australia (Non-Indigenous) ^13^	2005–2011	--	2011–	Y	92	92	PPV, PCV	PPV, PCV
	3 + 1/2 + 1	LA and C	Venezuela (SIREVA)	--	--	2014–	Unk	--	7	--	PPV, PCV
		Europe	Germany ^10^	2006–2009	--	2009–	N	85	84	PPV	PPV, PCV
			Spain, Catalonia	2001–2010 ^9^	--	2010–2015 ^9^ 2016–	N	70 ^9^	93	PPV	PPV, PCV
			Spain, Navarra	2004–2009 ^9^	--	2010–2015 ^9^ 2016–	N	71^9^	93	PPV	PPV, PCV
PCV10/13	2 + 1	N. Am.	Canada, Quebec (excluding Nunavik)	2004–2009	2009–2010	2011–2018	N	97	75	PPV	PPV, PCV
					2018–						
		LA and C	Chile, Metropolitan Region	2009–2010	2011–2015	2016–	N	97	89	PPV	PPV
			Chile, Non-Metropolitan Regions	--	2011–2017	2017–	N	97	89	PPV	PPV
			El Salvador (SIREVA, WHO)	2010–2011	2018–	2011–2018	Unk	--	87	PPV	PPV, PCV
			Paraguay	--	2012–2017	2017–	Y	78	91	PPV	--
			Peru (SIREVA, WHO)	2009–2011	2011–2015	2015–	N	--	--	PPV	PPV
		Europe	Belgium	2007–2011	2015–2019	2011–2015	N	93^8^	94	PPV, PCV	PPV, PCV
						2019–					
			Slovakia	2009–2010	2011–	2011–	Y	97	97	PCV	PCV
			Sweden	2009–2010	2010–	2010–2019	N	97 ^8^	97	PPV	PPV, PCV
		N. Africa and W. Asia	Morocco, Grand Casablanca	--	2012–	2010–2012	N	91	90	--	--
	3 + 1	N. Am.	Canada, Quebec-Nunavik	2002–2009	2009–2010	2011–	N	97	75	PPV	PPV, PCV
			Canada, Ontario	2005–2009	2009–2010	2010–	Y	72 ^8^	77	PPV	PPV, PCV
		Asia	Hong Kong	2009–2010	2010–2011	2011–	N	98	--	PPV, PCV	PPV, PCV
		Oceania	New Zealand	2008–2011	2011–2014, 2017–	2014-2017	N	93	93	PPV, PCV	PPV, PCV
			Australia, Northern Territory	2001–2009	2009–2011	2011–	Y	88	92	PPV, PCV	PPV, PCV
	3 + 1/2 + 1	Europe	Czech Republic	--	2010–	2010–	N	74 ^8^	--	PPV, PCV	PPV, PCV
			Poland ^11^	--	2017–	2017–	N	94	60	PPV, PCV	PPV, PCV

^1^ Product and schedule classifications intend to represent what was widely used in the population as of 2018, which occasionally differ from the national universal recommendation. ^2^ United Nations (UN) regions adapted from UN Statistics Division [25]. ^3^ (WHO): WHO Global Invasive Bacterial Vaccine-Preventable Diseases (IB-VPD) Surveillance Network; (SIREVA): Pan American Health Organization Sistema de Redes de Vigilancia de los Agentes Responsables de Neumonias y Meningitis Bacterianas (SIREVA); (ECDC): The European Surveillance System (ECDC). ^4^ “—” represents PCV not universally used. ^5^ Annual PCV uptake estimates provided by the surveillance site for the primary series of PCV by 12 months of age (if available, for some sites up to 15 months of age), excluding year of vaccine rollout; ‘—’ represents no coverage information provided to PSERENADE project. ^6^ WUENIC PCV3 uptake, excluding the year of vaccine rollout (PCV3 represents the third dose whether given before 12 months or at or after 12 months, but in some cases uptake estimates may reflect the percentage of surviving infants who received two doses of PCV prior to the 1st birthday); ‘—’ represents no WUENIC coverage information available. ^7^ Pneumococcal vaccine recommendation for other age groups or risk conditions. Adult recommendations are for all adults aged 50 years and above, aged 60 years and above, or aged 65 years and above. High-risk population age recommendations and populations included varies across sites. ‘--’ represents no product is recommended. ^8^ Annual PCV uptake estimates provided by the surveillance site for the primary series plus the booster dose by 23 months of age, excluding year of vaccine rollout. ^9^ Recommended for high-risk populations only but had substantial (≥50% annually) private market uptake among the general population. ^10^ Although both PCV10 and PCV13 were recommended in the guidelines, the country was classified according to the product that was in substantially wide use. ^11^ PCV10 and PCV13 became available in the private market in 2009 and 2010, respectively. Widespread use began in 2017 when PCV was introduced into the Polish National Immunization Program (NIP). Private market use of PCV uses a different schedule (3 + 1) than the NIP (2 + 1). To date, PCV10 has been chosen for the NIP and private market PCV13 use is approximately 30%. ^12^ Range of vaccine uptake is 49–60%. ^13^ Australia (non-indigenous) switched from 3 + 0 to 2 + 1 in 2018. Abbreviations: N. Am.: North America; LA and C: Latin America and the Caribbean; N. Africa and W. Asia: Northern Africa and Western Asia; Y: Yes; N: No; Unk: unknown; PCV: pneumococcal conjugate vaccine; PPV: pneumococcal polysaccharide vaccine; 3 + 1: 3 primary doses plus booster; 2 + 1: 2 primary doses plus booster; 3 + 0: 3 primary doses and no booster.

## Data Availability

Restrictions apply to the availability of these data. Data were obtained under data sharing agreements from contributing surveillance sites and can only be shared by contributing organizations with their permission.

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
