# Peer review of "Global Landscape Review of Serotype-Specific Invasive Pneumococcal Disease Surveillance among Countries Using PCV10/13: The Pneumococcal Serotype Replacement and Distribution Estimation (PSERENADE) Project"

_microorganisms, 2021, doi:10.3390/microorganisms9040742_

Round 1

Reviewer 1 Report

This article by Knoll et al. provides an overview of the methods and search process for the PSERENADE project. The authors should be commended for undertaking such a comprehensive and important project that will be critical in inform future directions with regards to pneumococcal vaccination policy, research, and implementation. There are some general questions about methodology, of which some may be addressed in future manuscripts as they may be specific to each research question of interest. There are also a few minor suggestions or points of clarification that are requested.

General Comments:

  • The specific reason for ineligibility of a certain region is not apparent. As one example, I see Papua New Guinea is not eligible, but they have surveillance systems, coverage data, and have run head-to-head trials of PCV10 and PCV13. I assume this valuable data is not included because national level coverage is under 50% (although the largely isolated regions where the studies were carried out may be higher).
  • Similarly, it is not readily apparent what proportion of data was excluded for each of the data quality checks.
  • Table 2: Data quality check B – I agree with the overall approach, however unless I am misunderstanding, excluding non-decreasing years or datasets may bias certain evaluations or reduce the capacity to evaluate patterns of replacement, especially if specific products are not as effective against certain serotypes in specific settings (for example, ST19A and ST3 in the UK, and ST1 in certain settings). It seems that elucidating associations driving replacement would be of particular interest and excluding data where this may be occurring would make this more challenging.
  • Methods, line 279 and Table 2: These are excellent data quality checks. Was the implementation of this a qualitative discussion with the site, or were there specific quantitative thresholds? For example, what constitutes a “dramatic change”? This wording is potentially subjective – a 20% change in a large well-established surveillance system may be flagged, but a 20% change in a smaller relatively noisy system may not be identified. If so, would that lead to potential biases for being more likely to include data from larger, more stable datasets?
  • Although this may not be within the scope of this paper, the issues raised about gaps in the data and how the majority of data were from HICs whereas the burden of disease is highest in LMICs raises questions about generalizability of findings and potential solutions for analytic weighting. I assume the handling of the data weighting, stratification, and generalizability will be described in the context of individual analyses coming out of the project but were prominent questions while reading through the paper.

Minor Comments:

  • Abstract: I may be misinterpreting, but the numbers of sites using each product and the number for different dosing schedules both seem to total to more than the number of countries in PSERENADE (n=63).
  • Introduction, line 169: The paper states that “Even sites not affected by these issues cannot assess the relative merits across PCV products or schedules…”. Many included sites have run comparisons of different vaccine products or comparisons of schedules. These within-site controlled trials seem to be valuable in informing relative merits while controlling for setting.
  • Methods, line 198: Although mentioned in the abstract, the reader may not know the timeframe for when PSERENADE was conducted, so the “by 1 January 2017” may be more helpful later in the methods section.
  • Methods, line 220: Did the study include studies written in any language? I believe many prior evaluations of PCV and serotype replacement were done with only English articles so this may be a notable change.
  • Table 1: Is the 50% uptake requirement at the national level, or the region covered by the surveillance system? If the former, were studies without national level coverage data excluded even if there was robust coverage and data from a large regional surveillance system?
  • Methods, line 273: Missing “by” in “adjudicated a third reviewer”.
  • Table 4: It is not immediately clear what the n represents in the table (studies vs. sites, vs. countries).
  • Results, line 420: I may be misinterpreting, but the percents do not seem to add up, unless both PPV and PCV are recommended, in which case the wording could be adjusted.
  • Some references have a format I have not seen previously with regards to the journal name. For example, in my experience CID does not usually have the “Official publication of the Infectious Diseases Society of America” tag when referenced.
  • Supplemental tables: Table of contents is not showing page numbers for some links.

Author Response

Reviewer #1

  1. This article by Knoll et al. provides an overview of the methods and search process for the PSERENADE project. The authors should be commended for undertaking such a comprehensive and important project that will be critical in inform future directions with regards to pneumococcal vaccination policy, research, and implementation. There are some general questions about methodology, of which some may be addressed in future manuscripts as they may be specific to each research question of interest. There are also a few minor suggestions or points of clarification that are requested.

We thank the reviewer for pointing out that the project will be important for pneumococcal vaccine policy and for their insightful comments which we believe have improved the quality of the manuscript. We have tried our best to improve the manuscript by addressing the reviewer’s comments.

  1. The specific reason for ineligibility of a certain region is not apparent. As one example, I see Papua New Guinea is not eligible, but they have surveillance systems, coverage data, and have run head-to-head trials of PCV10 and PCV13. I assume this valuable data is not included because national level coverage is under 50% (although the largely isolated regions where the studies were carried out may be higher).

We are not interested in the efficacy of the products and so we did not include RCTs. Additionally, clinical trial data are not appropriate for this work because they can't inform on the indirect effects of the vaccination program (which includes herd immunity as well as serotype replacement), nor can they inform on the long-term effects of a PCV program. IPD surveillance years with less than 50% immunization coverage were excluded because we are interested in the effectiveness of robust programs in order to answer the question, “if PCV10/13 is used with high (i.e., >=70%) coverage, what should countries expect to see with respect to impact on IPD?”. We want any heterogeneity in the results to not reflect coverage but rather the potential of PCV itself.

  1. Similarly, it is not readily apparent what proportion of data was excluded for each of the data quality checks.

The data quality checks were used to make inclusion decisions for specific analyses and not for data collection. The proportion of data excluded is/ will be described in detail in manuscripts on each specific analysis. We have better data on the quality for those for whom we requested the raw data. More details on sites’ data quality is described in manuscripts on specific analyses because eligibility is analysis specific, meaning some quality issues may be irrelevant for one analysis but relevant for another, or we choose different thresholds for a quality criterion depending on the analysis. For sites for which data were not received we were unable to evaluate data quality. The proportion of data meeting data collection eligibility is somewhat explained in Table 3 by seeing who remains as you move column to column.

  1. Table 2: Data quality check B – I agree with the overall approach, however unless I am misunderstanding, excluding non-decreasing years or datasets may bias certain evaluations or reduce the capacity to evaluate patterns of replacement, especially if specific products are not as effective against certain serotypes in specific settings (for example, ST19A and ST3 in the UK, and ST1 in certain settings). It seems that elucidating associations driving replacement would be of particular interest and excluding data where this may be occurring would make this more challenging.

We understand that this may appear circular. We were very careful to evaluate decreasing data for vaccine serotypes where declines should have been seen (i.e., in the years immediately following vaccine introduction in the target age group). For example, because clinical trial data show high efficacy of PCV7 against PCV7 stereotypes, if we did not see declines in PCV7 serotypes + serotype 6A in children, this raised suspicions about the coverage and/or the surveillance system. We also expected to see declines in serotypes 1, 5, and 7F following PCV10 or PCV13 introduction and declines in serotypes 19A following PCV13 introduction. The impact of serotype replacement, for example of 19A during the PCV7 use period, was considered. If serotype 3 did not decline following PCV13 introduction this was noted but was not considered to be unexpected given PCV13 efficacy data against serotype 3. These identifications resulted in conversations with the sites to better understand the patterns observed in the data and the exclusion of certain years from specific analyses was determined in collaboration with and with approval from the sites.

  1. Methods, line 279 and Table 2: These are excellent data quality checks. Was the implementation of this a qualitative discussion with the site, or were there specific quantitative thresholds? For example, what constitutes a “dramatic change”? This wording is potentially subjective – a 20% change in a large well-established surveillance system may be flagged, but a 20% change in a smaller relatively noisy system may not be identified. If so, would that lead to potential biases for being more likely to include data from larger, more stable datasets?

Implementation of data quality checks was a conversation with the sites at the qualitative level aimed to understand changes in either the surveillance system or challenges with the rollout of the vaccine in the early years. These conversations were broad and encompassed a variety of elements, including private market PCV use, changes in reporting of IPD cases, and changes in serotyping methods. We did not apply a single threshold to all sites. Rather, visual changes of sufficient magnitude to meaningfully bias the interpretation of the data were discussed with the site. Any years of data where the site agreed there were meaningful changes in surveillance or poor coverage were excluded as appropriate for specific analyses. In addition, the sample size was considered. We looked at both the absolute case counts as well as incidence rates to identify patterns of concern and to consider randomness due to small sample size as an explanation. We have added ‘… and discussions with investigators at each site…’ to the Methods section in lines 281-282.

  1. Although this may not be within the scope of this paper, the issues raised about gaps in the data and how the majority of data were from HICs whereas the burden of disease is highest in LMICs raises questions about generalizability of findings and potential solutions for analytic weighting. I assume the handling of the data weighting, stratification, and generalizability will be described in the context of individual analyses coming out of the project but were prominent questions while reading through the paper.

We agree that this is a limitation not just for our database but in the amount and quality of information available globally. You are correct that this is addressed in the individual analysis papers. Regarding the comment of weighting, those analysis don't attempt to weight by geographic representation or to infer that the meta-estimates reflect the global picture (although there is the typical weighting by sample size and amount of heterogeneity in meta-estimates). This is because of the heterogeneity we observed, and because the amount of available data in LMICs are insufficient for confidently being able to address the imbalance seen with weighting (i.e., the 95% CI of any weights would be too wide). But we do try to address this by going to great lengths to describe the individual country data, to describe the heterogeneity, and to not label the aggregated (meta) estimates as representative of the ‘global’ picture. It is good to know, though, that despite heterogeneity on some aspects (i.e., indirect effects on adults), there are a lot of analyses showing homogenous responses (e.g., net effects on all IPD in children <5y) that do allow us to make generalized statements that are supported by individual country data across varied settings and conditions.

  1. Abstract: I may be misinterpreting, but the numbers of sites using each product and the number for different dosing schedules both seem to total to more than the number of countries in PSERENADE (n=63).

This is correct. There are some countries with more than one site and where PCV use may vary by region within the country. For example, Kenya and Spain each have data from three different sites. These are important to keep separate (as opposed to pooling within a country) because they differed in how vaccine was introduced (for example, PCV was rolled out in different years or PCV was introduced with catch-up vs without) and/or in their surveillance methods. We have noticed some places in tables where we could be clearer and have added footnotes.

  1. Introduction, line 169: The paper states that “Even sites not affected by these issues cannot assess the relative merits across PCV products or schedules…”. Many included sites have run comparisons of different vaccine products or comparisons of schedules. These within-site controlled trials seem to be valuable in informing relative merits while controlling for setting.

Thank you. You raise a good point that this is not all or none and some countries have been able to evaluate comparisons of products or schedules for direct effects in children. The evaluation of effects of long term use and indirect effects in both children and older unvaccinated age groups is not possible though at this time for any country to have done as it would require use of PCV10 for approximately seven years and then a switch to PCV13 for approximately seven years (that is about what it would take to compare the indirect effects and have enough time for washout of the previous vaccine use). We have revised this sentence in the Introduction section in lines 169-170.

  1. Methods, line 198: Although mentioned in the abstract, the reader may not know the timeframe for when PSERENADE was conducted, so the “by 1 January 2017” may be more helpful later in the methods section.

We looked to see if we could find a more appropriate place, but because January 2017 is in reference to data collection (i.e., when we received the datasets from sites) we felt this should come early in the Methods section rather than later as it is the first step.  The timeframe is mentioned in three places: Abstract, Methods, and Results (line 300) so we hope that readers looking for this will be able to find it.

  1. Methods, line 220: Did the study include studies written in any language? I believe many prior evaluations of PCV and serotype replacement were done with only English articles so this may be a notable change.

Yes, the literature search included studies written in any language. We added this to the Methods section in lines 212. Also see lines 221-222 of the Methods section (‘Two reviewers fluent in the language of the written report…’). We did not let language be a barrier in searching for articles as we reached out to the community at broad to ask of any known data, including unpublished data. Most of the data were raw datasets provided by the countries including quite a lot of unpublished data, in which case there were no language issues to contend with. Communication with sites was sometimes conducted by a team member fluent in Spanish or French as appropriate, but countries with other languages always had an English speaker to work with us.

  1. Table 1: Is the 50% uptake requirement at the national level, or the region covered by the surveillance system? If the former, were studies without national level coverage data excluded even if there was robust coverage and data from a large regional surveillance system?

Where data were available, we attempted as best we could to match the uptake requirement to the population covered by the surveillance system. For some countries, good local data were not available and WEUNIC estimates were required to be used, which are notoriously poor estimates of coverage in some places as they can rely on poor-quality administrative data. So, we acknowledge that there may be some misclassification of this requirement at some sites (erring in overestimating coverage). However, we did attempt to confirm that the coverage was reflective of the surveillance population and we discussed this issue with the sites carefully before excluding any years of data. We think that where there is misclassification it is erroneously including years of data where coverage may have been lower than desired (i.e., relying on WEUNIC for lack of better data), but this is in very few sites. Please refer to lines 262-263 of the Methods section (‘…annual immunization uptake estimates representative of the population under surveillance…’).

  1. Methods, line 273: Missing “by” in “adjudicated a third reviewer”.

Thank you, we have made this edit.

  1. Table 4: It is not immediately clear what the n represents in the table (studies vs. sites, vs. countries).

Thank you, we have provided a description in the footnotes of Table 4 (lines 386-387).

  1. Results, line 420: I may be misinterpreting, but the precents do not seem to add up, unless both PPV and PCV are recommended, in which case the wording could be adjusted.

This is PPV23 and/or PCV13, as some countries have recommendations for both. This has been clarified in lines 422-424 of the Results section.

  1. Some references have a format I have not seen previously with regards to the journal name. For example, in my experience CID does not usually have the “Official publication of the Infectious Diseases Society of America” tag when referenced.

Thank you, we have problems with the auto function of the reference manager software we are using and have to manually edit this in the manuscript. We did not catch this one but will now add this to our list of checks for all manuscripts.

  1. Supplemental tables: Table of contents is not showing page numbers for some links.

Thank you, we have fixed this.

Reviewer 2 Report

Overview: This manuscript describes the global dataset that has been compiled for a WHO commissioned investigation into invasive pneumococcal disease incidence and serotype distribution among children and adults in the PCV10/PCV13 vaccine era. Limitations and gaps in the IPD datasets are identified in this study. Study sites are also described in detail. This manuscript is the foundation to future manuscripts that will assess PCV10/PCV13 impact on IPD within a global context. Five methods were used to compile this dataset: 1) contacting known surveillance networks, 2) comprehensive literature searches and 3) review of results from previous PCV reviews, 4) review of citations from post-PCV7 era, and 5) review of ISPPD abstracts from 2012-2018.

Strengths: The breadth of this dataset is impressive, with in-depth investigation ensuring that as many eligible datasets were included to provide a valuable multi-site database. This database will enable robust and more accurate analysis of IPD trends and PCV10/13 impact on IPD, which will inform vaccine policy on a global level.

MAJOR comment: The decision to exclude study sites that had changes in their IPD surveillance system over the analysis period (Table 1) is an interesting one that warrants some discussion. How many sites had a change in their surveillance system during the data collection period? Should these sites be included in sub-analysis?

MINOR COMMENTS:

Formatting in Table 1 and Table 2 needs to be changed so that the column text do not run into each other, currently it is very difficult to read.

Formatting is annoying throughout text, with many words hyphenated at right-hand side of page – this may be a journal formatting issue from upload rather than author issue? Please check.

Line 142: change ‘39 2+1, 13 3+1, and 8 mixed’ to ’39 had 2+1, 13 had 3+1, and 8 had mixed schedules’

Lines 151-154, consider splitting into two sentences i.e.: ‘In 2007, the World Health Organization (WHO) first recommended including pneumococcal conjugate vaccines (PCV) in childhood immunization programs worldwide to prevent pneumococcal disease. WHO encouraged countries to implement surveillance of invasive pneumococcal disease (IPD) to establish a baseline rate of disease for evaluating vaccine impact’.

Line 301: suggest removing ‘(Table 3)’ from this sentence ‘Characteristics associated with participation were not evaluated (Table 3), if they were not evaluated. Alternatively, this sentence could be changed to ‘Characteristics associated with participation were not evaluated but the proportion of participating eligible sites are detailed for each region (Table 3)’

Check table 3, column 9 for formatting of numbers: sometimes there’s a comma in the numbers e.g. 183,610, sometimes not e.g. 152977

Line 398: remove 2nd ‘sites’ from ‘surveillance sites (77.3%) sites’

Line 412: ‘Data on adult PPV23 and PCV13 uptake were largely unavailable but, where estimated, were low (data not shown).’ What does ‘low’ uptake mean? This statement would have more value if uptake rates were able to be given for sites that data was collected for. Is this possible?

Line 447 and 495: change ‘Further’ to ‘furthermore’

Line 486: Streptococcus, change to S.

Author Response

  1. Overview: This manuscript describes the global dataset that has been compiled for a WHO commissioned investigation into invasive pneumococcal disease incidence and serotype distribution among children and adults in the PCV10/PCV13 vaccine era. Limitations and gaps in the IPD datasets are identified in this study. Study sites are also described in detail. This manuscript is the foundation to future manuscripts that will assess PCV10/PCV13 impact on IPD within a global context. Five methods were used to compile this dataset: 1) contacting known surveillance networks, 2) comprehensive literature searches and 3) review of results from previous PCV reviews, 4) review of citations from post-PCV7 era, and 5) review of ISPPD abstracts from 2012-2018.

Strengths: The breadth of this dataset is impressive, with in-depth investigation ensuring that as many eligible datasets were included to provide a valuable multi-site database. This database will enable robust and more accurate analysis of IPD trends and PCV10/13 impact on IPD, which will inform vaccine policy on a global level.

We thank the reviewer for pointing out that this study was conducted with care and will be of interest for pneumococcal vaccine policy decisions. We have tried our best to answer the items raised in order to improve the manuscript.

  1. The decision to exclude study sites that had changes in their IPD surveillance system over the analysis period (Table 1) is an interesting one that warrants some discussion. How many sites had a change in their surveillance system during the data collection period? Should these sites be included in sub-analysis?

During the data request process, some sites communicated that their site did not meet data collection eligibility criteria, and therefore we did not move forward with requesting data for the project. We do not know individual reasons for exclusion for these sites that deemed their data ineligible in this first step which determined if it was worth the effort to send to us to evaluate further. We asked about a list of critical issues and if they deemed that they ‘failed’ any they did not report the reason why. Therefore, we cannot determine the number that failed on surveillance system changes.

However, no sites who sent data were completely excluded due to changes in the surveillance system over time. For some sites, specific years or specific age groups were excluded from certain analyses depending on analytic eligibility criteria relevant to each analysis (as not all criteria pertain to all analyses). Therefore, only those that are relevant are applied to each analysis. So, site data may be used in some analyses and are excluded in other analyses. This is/ will be described in detail in manuscripts for each analysis. For those analyses that excluded data because they will likely produce biased estimates (i.e., serotyping more cases in the post-PCV period relative to the pre-PCV period which produces an apparent increase in serotyped incidence), we will not run sensitivity analyses to include them because it does not help us understand the true effects of PCV better, but only introduces bias. However, for some of these issues, we try to handle by adjustment (for that example, we can adjust for % serotyped and thus include the data). Each exclusion/adjustment (whether for specific years of data or entire site) has been discussed and agreed upon by the Technical Advisory Group and the site in question as being the decision that produces the most unbiased results.

  1. Formatting in Table 1 and Table 2 needs to be changed so that the column text do not run into each other, currently it is very difficult to read.

Thank you. When the journal formats the tables, we will check that this is taken care of.

  1. Formatting is annoying throughout text, with many words hyphenated at right-hand side of page – this may be a journal formatting issue from upload rather than author issue? Please check.

We agree. It is a template issue that the Journal submission software produces. I do not think we have control over the look of the formatting which the Journal will do. However, hopefully this will improve at the proofs stage.

  1. Line 142: change ‘39 2+1, 13 3+1, and 8 mixed’ to ’39 had 2+1, 13 had 3+1, and 8 had mixed schedules’

Thank you, we have made this change.

  1. Lines 151-154, consider splitting into two sentences i.e.: ‘In 2007, the World Health Organization (WHO) first recommended including pneumococcal conjugate vaccines (PCV) in childhood immunization programs worldwide to prevent pneumococcal disease. WHO encouraged countries to implement surveillance of invasive pneumococcal disease (IPD) to establish a baseline rate of disease for evaluating vaccine impact’.

Thank you, we have made this change.

  1. Line 301: suggest removing ‘(Table 3)’ from this sentence ‘Characteristics associated with participation were not evaluated (Table 3), if they were not evaluated. Alternatively, this sentence could be changed to ‘Characteristics associated with participation were not evaluated but the proportion of participating eligible sites are detailed for each region (Table 3)’

Thank you, we have made this change.

  1. Check table 3, column 9 for formatting of numbers: sometimes there’s a comma in the numbers e.g. 183,610, sometimes not e.g. 152977

Thank you. When the journal formats the table we will check that this is taken care of.

  1. Line 398: remove 2nd ‘sites’ from ‘surveillance sites (77.3%) sites’

This has been corrected.

  1. Line 412: ‘Data on adult PPV23 and PCV13 uptake were largely unavailable but, where estimated, were low (data not shown).’ What does ‘low’ uptake mean? This statement would have more value if uptake rates were able to be given for sites that data was collected for. Is this possible?

Thank you for raising this point. We have added this summary in lines 426-428 of the Results section.

  1. Line 447 and 495: change ‘Further’ to ‘furthermore’

We have made this change.

  1. Line 486: Streptococcus, change to S.

We have made this change.

Round 2

Reviewer 1 Report

The authors have responded to all concerns and questions thoroughly and I approve of all the modifications and revisions. The coverage information provided in the responses was helpful to me and the authors may consider whether some of that information would help a general audience and included in the methods or discussion; specifically, the use of WEUNIC estimates when local data was not available and potential misclassification may be informative (i.e. overestimation of coverage).